# Experimental Evidence of Bone Lesions Due to Larder Beetle *Dermestes maculatus* (Coleoptera: Dermestidae)

**DOI:** 10.3390/biology11091321

**Published:** 2022-09-06

**Authors:** Damien Charabidzé, Vincent Lavieille, Thomas Colard

**Affiliations:** 1University of Lille, CNRS, Centre d’Histoire Judiciaire, UMR 8025, F-59000 Lille, France; 2Université Libre de Bruxelles (ULB), Unit of Social Ecology (USE), B-1000 Bruxelles, Belgium; 3University of Bordeaux, CNRS, MCC, PACEA, UMR 5199, F-33600 Pessac, France; 4University of Lille, CHU Lille, Department of Oral and Maxillofacial Radiology, F-59000 Lille, France

**Keywords:** larder beetles, forensic entomology, taphonomy, X-ray CBCT, forensic anthropology

## Abstract

**Simple Summary:**

Larvae of the larder beetle *Dermestes maculatus* (Coleoptera: Dermestidae, De Geer, 1774) resemble caterpillars but actually feed on dry tissues of dead animals, and sometimes on human corpses. To hide and metamorphose into adult beetles, these larvae dig deep tunnels called pupation chambers. These holes and pits are usually observed in skins, horns, furs, or surrounding materials such as wood. They have also been reported on archaeological bones and experimentally observed on fresh pig bones, but never on humans. In this context, we investigated whether larder beetle larvae could also dig pupation chambers in human bones, and under which conditions this could occur. For this purpose, we placed medieval dry human bones as well as fresh calf and beef bones (control) with *Dermestes maculatus* larvae. After 1 month, we observed tunnels corresponding to pupation chambers only on dry human bones, and under conditions of high larval density. Despite these results being preliminary, they are, nonetheless, of particular interest in a forensic context, as they could help to understand taphonomic bone modifications or even the chronology of mass grave deposals.

**Abstract:**

Dermestid beetles (Coleoptera: Dermestidae) are necrophagous insects feeding on mummified carcasses. After six to seven molts, the larvae stop feeding and dig pupation chambers to hide and safely evolve into adults. Such pupation chambers have already been observed on archaeological mammals’ bones, but the attribution and interpretation of these osteological lesions lack experimental evidence in a forensic context. To observe whether dermestid larvae dig pupation chambers in human bones, 20 or 40 *Dermestes maculatus* (De Geer, 1774) larvae were placed in a dermestarium with different types of bones varying in species (*Bos taurus* or human), age (adult or immature), and preservation method (fresh or dry). Our results show that dermestid larvae caused multiple lesions, including larval mandible traces on cortical bone, cortical perforations, drilling of pupation chambers, destruction of the trabecular network, and the perforation of cartilage. Bone destruction was mainly observed on aged dry bones, while fresh bones only exhibited soft tissue and superficial cartilage lesions. According to these results, pupation chambers could indicate the simultaneous presence of several corpses at different decomposition stages, or the addition of new corpses while others were already skeletonized. These conclusions are particularly important in the case of mass graves, where chronology is sometimes difficult to establish.

## 1. Introduction

Dermestid beetles (Coleoptera: Dermestidae), also known as larder beetles, are necrophagous insects feeding on dry organic material, including human corpses. These insects are fairly common, and were initially known as pests of stored products and for their use in natural history museums to clean skeletal remains. In a forensic context, they are often observed on human corpses during late decomposition stages, but have also been sampled as early as 10 days after death [1,2,3]. During colonization, larder beetles accelerate the skeletonization process at varying rates depending on population composition and density, body size, ambient temperature, or the amount of food available [4,5,6]. They are found in indoor as well as outdoor environments, especially in areas with dry weather or during warm months with low rainfall [7]. They have a worldwide distribution, with eight species reported from human corpses in northwestern Europe: *D. frischii, D. undulatus, D. peruvianus, D. lardarius, D. haemorrhoidalis, D. maculatus, D. bicolor*, and *D. ater* [1].

Larder beetles are holometabolous insects, which means they have four main stages during their life cycle: egg, larval, nymph, and adult. After mating, the females lay tens to hundreds of eggs in cracks or cavities of the cadavers. The larvae feed on dry flesh: they bore holes of 2–5 mm in diameter into desiccated muscle tissue, thus progressively creating an irregular network [3]. During this process, their mandibles can carve grooves in the cartilage and, sometimes, on the edges of bones [8]. After six to seven molts, larvae reach a sufficient weight for metamorphosis [9]. They stop feeding and dig pupation chambers to hide and safely evolve into nymphs and adults. The duration of this development cycle depends on temperature, but also on food moistening and ambient hygrometry [10,11]. Several generations can follow, as long as food is available [12].

Due to their ability to remove the remains of flesh from bones, taxidermists and forensic anthropologists have widely used dermestid larvae to deflesh skeletons [4,13,14]. However, larvae have also been reported to sometimes cause osteological lesions ranging from grooves to furrows [15,16,17,18]. However, osteophagy (i.e., the ingestion of bone material as food) probably occurs scarcely, and only in situations with a shortage of other dry organic tissues [8]. Most of the lesions that are observed are most likely due to larvae digging pupation chambers directly into bones [8,12,19].

Indeed, just before nymphosis, dermestid beetle larvae dig tunnels called pupal chambers to hide and reduce the risk of predation as well as cannibalism by other larvae [20,21,22]. The formation process and shaping of these holes have been described in detail by several authors, mostly in the archaeological literature [16,17,18]. While they are usually observed in skin, horns, dry tissues, or surrounding materials such as wood, chambers can sometimes occur in animal bones [8,15,16]. Most authors, however, agree that larvae use bones as places to hide only under exceptionally harsh conditions, notably when a large concentration of larvae must cope with little food [12,23]. Furthermore, there is no case report or experimental evidence of larder beetle pupal chambers in human bones so far. Thus, the interpretation of osteological lesion formation as a consequence of dermestid beetles lacks experimental evidence in a forensic context.

Along with an a posteriori analysis, only a handful of experimental studies have been performed so far: Huchet et al. (2013) found pupation chambers on polystyrene and wood, while Zanetti et al. (2014, 2019) reported them on pig bones [16,23,24]. However, it is still unclear whether dermestid beetles can actually dig pupation chambers in adult human bones, and whether fresh skeletons can be altered as well as dry ones. These questions are fundamental in forensic investigations when working on cold cases or trying to determine the taphonomy of a skeleton. In this context, the present study aimed to test the ability of dermestid larvae to dig pupal chambers in human bones and determine the conditions favoring this behavior.

## 2. Materials and Methods

*Dermestes maculatus* (Coleoptera: Dermestidae) (De Geer, 1774) is one of the most synanthropic dermestid species, and is commonly found in forensic cases in Europe [1,3,7,24,25]. Specimens used for experiments came from a main colony mixing individuals from three different laboratory colonies: Frankfurt University (Frankfurt, Land of Hess, Germany), Simon Fraser University (Burnaby, BC, Canada), and Bauer Handels GmbH company (Fehraltorf, Switzerland). For experimental purposes, 240 larvae measuring between 7 and 12 mm in length and 3–4 mm in width (i.e., larval stages 4 to 6) were sampled from the main colony [9].

To observe whether dermestid larvae dig pupation chambers in bones, 20 or 40 larvae were placed in a dermestarium with food and different types of bones. The different conditions detailed below were designed to observe the effect of bone freshness as well as the link between larval density (resulting in a cannibalistic pressure) and the digging of pupal chambers [24,26,27].

### 2.1. Experimental Setup

The experiments took place in polycarbonate boxes supplied with dry food ad libitum (chicken-based cat pellets, Frolic©) and damped cotton [14,28]. Boxes were kept at 25 ± 2 °C and 50 ± 15% RH. As dermestid beetles avoid bright light and quickly disperse when stressed, all the colonies were kept in the dark without any intervention or disturbance. The experiments were stopped after 30 days, when moving larvae were no longer visible (i.e., all living larvae reached the nymph stage) [4]. To test the effect of larval density, 20 larvae experiments took place in 28.5 × 14.5 × 12 cm boxes (corresponding to a ground density of 0.04 larvae/cm^2^) while 40 larvae experiments were placed in 14 × 9 × 9.5 cm boxes (ground density of 0.31 larvae/cm^2^). This second condition artificially increased the average density of larvae, reproducing overcrowding situations.

Eight experiments were conducted simultaneously with three different types of bones varying in species (*Bos taurus* or human), age (adult or immature), and preservation method (fresh or dry) (Table 1). Each bone was placed randomly in a separate box, according to the conditions listed in Table 1. Fresh *Bos taurus* bones from a calf and an adult were purchased from a local butcher shop, defleshed, and cut into two parts: femoral head and distal epiphysis. The aged dry human bones were medieval archaeological remains (Bordeaux University collections) destined to be re-buried: one proximal extremity of a left adult femur, one right adult calcaneus, two adult vertebrae (lumbar, included in the same box), two fragments of an immature radius, and a proximal epiphysis of a right immature ulna (included in the same box).

### 2.2. Imaging

All bones were photographed before and after the experiments with a digital camera mounted on a 3D stand monitored with the Helicon Remote 3.8.6w software. Picture compilation was achieved with the Helicon focus 7 software and Adobe Photoshop CS4. Microscopic analysis was performed on a Nikon SMZ18 microscope (software NIS-Element BR) and a Leica M80 stereomicroscope (software LAS Core). After experiments, bone artifacts were scanned using Cone Beam Computerized Tomography (CS 93000 CBCT, Carestream^®^, Noisy-le-Grand, France), with a 15 × 10 cm field of view and a spatial resolution of 90 microns. Multi-planar reconstruction (3D-MPR) was performed using CS-3D Imaging software. The 3D analysis and post-treatment were performed using Avizo 9 (Thermo Fisher^®^, Bordeaux, France). Manual segmentation tools were used to calculate 3D volumes and surfaces. 

## 3. Results

Our results show that dermestid larvae caused multiple lesions on bone remains. These lesions displayed a large diversity, including larval mandible traces on cortical bone, cortical perforations, drilling of pupation chambers, destruction of the trabecular network, and perforation of cartilage. Bone destructions were mainly observed on aged dry bones: four of the seven dry bones used in this study exhibited bone damage, ranging from erosion to total perforation of the cortical bone. Extensive trabecular destruction was also observed on the calcaneus, the ulna, and the two radii. On the contrary, fresh bones only exhibited soft tissue and superficial cartilage lesions (Table 1).

### 3.1. Fresh Bones (Bos Taurus)

All the four fresh *Bos taurus* bones showed traces of dermestid activity. These lesions were only superficial, consisting of seven gnawing areas and eight cavities located in the decaying soft parts. All the eight cavities had a rounded, sometimes oval orifice, and their diameter decreased with depth, which corresponds to a pupation chamber dug from the outside to the inside. These superficial lesions were observed at both low and high larval density, and on juveniles as well as adult bones (i.e., all tested conditions). We also observed nymphs still present inside some of these cavities (Figure 1).

### 3.2. Human Calcaneus

The calcaneus showed two circular lesions located on its anterolateral articular facet (Figure 2A,B). The two perforations corresponded to entry holes that destroyed the cortical bone and penetrated deep into the trabecular bone. Scalloped margins surrounding the entry orifice (Figure 2C) were observed. The two holes were contiguous, but their paths was divergent (Figure 2D,E). The dimensions measured by X-ray imaging consisted of a maximum diameter of 4 mm and a depth of 5.8 mm on lesion I, and a maximum diameter of 5.9 mm and depth 7.5 mm on lesion II.

### 3.3. Human Ulna

The ulna displayed two circular perforations located on the lateral side of the proximal epiphysis: this area showed slight cortical erosion before the start of the experiment, exposing the trabecular bone (Figure 3A,B). Both lesions were similar to those observed on the calcaneus. The cortical surface was perforated and two tunnels diverging in trajectory had penetrated deep into the trabecular bone (Figure 3C,D). The dimensions comprised a maximum diameter of 5.4 mm with a depth of 13 mm on lesion I, and a max diameter of 4.7 mm and a depth 11.8 mm on lesion II.

CBCT imaging also showed evidence of nymphs and molts abandoned in situ after pupation, associated with trabecular bone destruction, attesting the presence of larvae within the medullary cavity (Figure 4).

### 3.4. Human Radius (Proximal Fragment)

The proximal fragment of the human radius 1 showed extensive lesion, with a large trabecular bone destruction and two cortical perforations observed in the proximal epiphysis (Figure 5). These two perforations did not lead to a pupal chamber as the entire trabecular bone was finally destroyed by larvae. The two perforations were different in shape: the larger one was circular with a maximum diameter of 4.5 mm, and the second was quadrangular with a size of 1.6 × 1.2 mm wide. CBCT slices and 3D rendered volumes showed a trabecular destruction and a thinning of the cortical inner surface (Figure 5C).

### 3.5. Human Radius (Distal Fragment)

The distal fragment of the human radius showed no visible lesions on the cortical surface. However, CBCT imaging revealed the presence of dermestid activity within the medullary cavity (Figure 6). Larvae had drilled several pupation chambers destroying trabecular bone and thinning the cortical inner surface. The dimensions measured by X-ray imaging were a diameter of 4.3 mm and a depth of 7.9 mm on lesion I, a diameter of 4.2 mm and a depth of 10.1 mm on lesion II, and a diameter of 4 mm and a depth of 6.4 mm on lesion III.

### 3.6. Dry Human Bones Placed at Low Larval Density

The three dry human bones (one adult femur proximal epiphysis and two lumbar vertebrae) placed at low larval density did not show any deep nor visible superficial lesions after experiments (Table 1).

## 4. Discussion

This study demonstrates, for the first time, the ability of *D. maculatus* larvae to create deep osteological lesions on human bones. Such entomological alterations are especially important in a forensic context, as they provide information on the history of the bone remains and the taphonomic processes involved in their degradation. However, it must be kept in mind that the results of this study are based on a limited set of experiments and need to be replicated. Thus, the following interpretations must be considered with precaution and regarded as hypothesis still needing more experimental confirmation.

The deep osteological lesions observed in the present study on human bones comply with the characteristics of pupal chambers previously established by Martin and West on other substrates (1995) [18]. Their diameter was slightly larger than the width of last stage larvae, and their narrowed entrance diameter resulted in an ovoid shape. These chambers started from the surface of cortical bone or directly through the medullary cavity, and extended into the trabecular bone. In some cases, successive drilling of the chambers destroyed the trabecular bone; over time, a bone used as a pupal site by many larvae may become hollow. In such a case, the attribution of bone lesions to larder beetles would be tricky and difficult to distinguish from others taphonomic processes, such as scavenging, weathering, root etching, or acidic water dissolution [29].

From the outside, the passage of dermestids can be misinterpreted as accidental injuries drilling into cortical bone. However, the scalloped margins on the edge of entry orifices, the thin scratches observed on cortical surface, and the pupation chambers drilled in the trabecular bone, proved to be the action of the mandibles of dermestid larvae [18]. In the absence of such distinctives patterns, the medullary cavity can present thin superficial pits that were previously the chamber’s bottom. This type of lesion is of major interest, as they can highlight the existence of pupal chambers even after the destruction of the trabecular bone. A detailed microscopic analysis of the bones’ surface, coupled with 3D imaging of the internal parts, is, however, necessary to observe such traces.

Pupal chambers were observed only on dry human bones exposed to high densities of larvae: none of the fresh bones nor those exposed to low larval density had such osteological lesions. However, many perforations were observed on dry flesh, some still containing nymphs in situ. According to these observations, it can be concluded that dermestid larvae preferentially dig pupation chambers into the soft desiccated flesh remaining on the surface. Larvae used dry bones as sites for pupation only if the predation pressure from conspecifics was high and soft mediums were not available in the surroundings.

These behavioral observations could seem anecdotal but may have important consequences in a forensic context [29]. However, the interpretations developed hereafter rely on a limited set of experiments and will need further confirmation.

First, our results indicate that pupation chambers were excavated only in dry bones, not fresh ones. This finding implies that bones with pupation chambers were already dry when pupation chambers were dug. However, larvae cannot feed on skeletonized remains and need food (flesh) to develop and reach the pupal stage: the existence of pupation chambers cannot be explained without the close proximity of flesh. Accordingly, the digging of pupation chambers directly into bones should occur on cadavers with asymmetric decomposition (some parts skeletonized while flesh is still presents on others). In this case, the topological distribution of the marks should be considered: chambers only dug on the limb could suggest a delayed conservation of the trunk, while chambers dug on the axial skeleton illustrate complete skeletonization of the corpse when chambers were dug. In this latter case, the presence of another corpse, not yet skeletonized, should therefore be necessary to explain the successful development of the larvae.

In the case of pupation chambers observed on an extensive set of bone remains or a whole skeleton, our results suggest the presence of several corpses at different decomposition stages, or the addition of new corpses while others were already skeletonized [30]. In such a situation, bones with pupation chambers should be considered to be the oldest, as they were already dry and served as refuges, while other corpses provided the flesh required for larval development. These conclusions are particularly important in the case of mass graves, where chronology is sometimes difficult to establish [31].

Finally, the occurrence of bone pupation chambers only under high larval density/overcrowded conditions could help to explain why these alterations are uncommon. Indeed, according to our results, the digging of pupation chambers directly into bones should occur only under certain conditions allowing the development of massive larval populations. Larder beetles and their larvae are most active during the hot and dry summer months, when rapid desiccation of carrion is more likely and larval development is fast [1,3]. Consequently, the presence of pupation chambers could indicate the seasonality and a narrow temperature range during the period of larval activity [18,30].

## 5. Conclusions

Larder beetles are used by natural history museums to clean skeletal remains. Their ability to quickly remove flesh from bones is also of interest to forensic anthropology laboratories. Indeed, the use of mechanical methods to remove soft tissues can damage the surface of the bones, and simmering with detergent or bleach is not DNA-safe. However, it is necessary to ensure that dermestid larvae do not create superficial or deep lesions during the cleaning process. Together with former studies, our results indicate that *D. maculatus* larvae bred at low density with food ad libitum do not alter fresh bones [4]. We do, however, recommend the use of young larvae, provided with a pupation substrate such as wood or polystyrene, and the removal of the bones before larvae start to pupate.

The interpretation of osteological lesions is of primary importance in forensic taphonomy to determine the cause of the death as well as the peri-mortem and post-mortem events. Information provided by insect activity has been underestimated so far. On the contrary, archaeological literature has long been interested in the impact of environmental factors, including insect artifacts, on bones. This study expands across these two fields to provide the first experimental evidence of larder beetle pupation chambers on human bones. While preliminary, these results could bring decisive information in the interpretation of mass graves.

## Figures and Tables

**Figure 1 biology-11-01321-f001:**
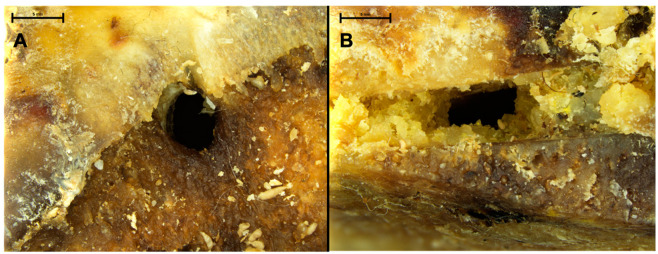
Close view of Bovidae fresh femoral head showing circular holes (**A**) and rectangular hole on a chop mark (**B**), both corresponding to pupation chambers dug into the decaying flesh and cartilage.

**Figure 2 biology-11-01321-f002:**
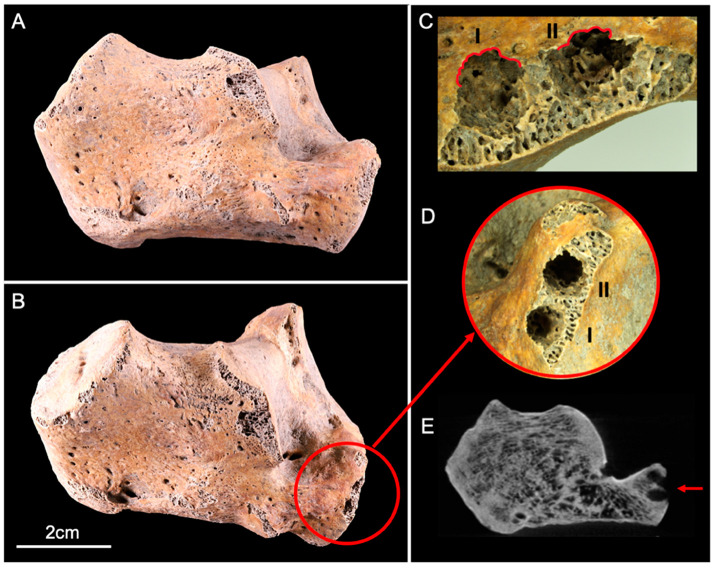
Lateral view of the human calcaneus before (**A**) and after (**B**) the experiment. The red circle (**B**,**D**) delineates two circular lesions (I and II) located on the anterolateral articular facet, with scalloped margins ((**C**), red lines). (**E**) CBCT cross-section shows the two lesions (red arrow) that destroyed the cortical and trabecular bone.

**Figure 3 biology-11-01321-f003:**
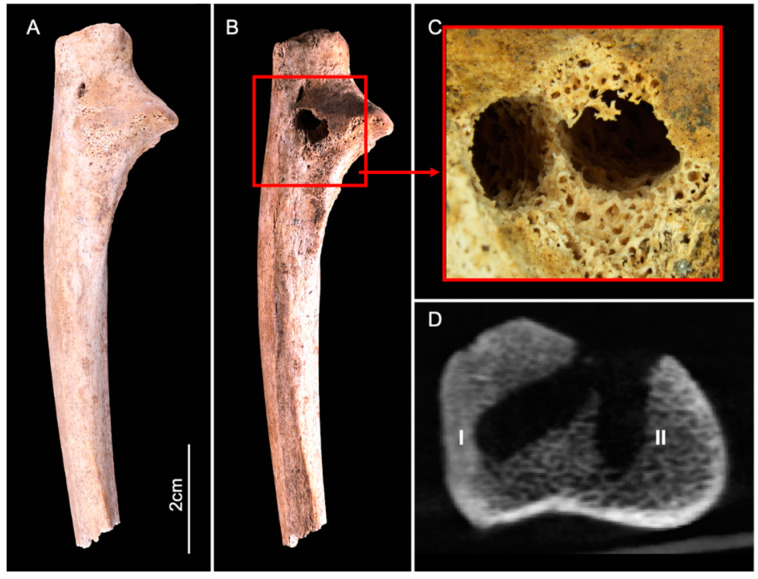
Lateral view of the human ulna before (**A**) and after (**B**) the experiment. The red square (**B**,**C**) delineates two adjacent circular lesions (I and II), located on the lateral side of the proximal epiphysis. (**D**) CBCT cross-section shows the two lesions that destroyed the cortical bone and penetrated deep into the trabecular bone.

**Figure 4 biology-11-01321-f004:**
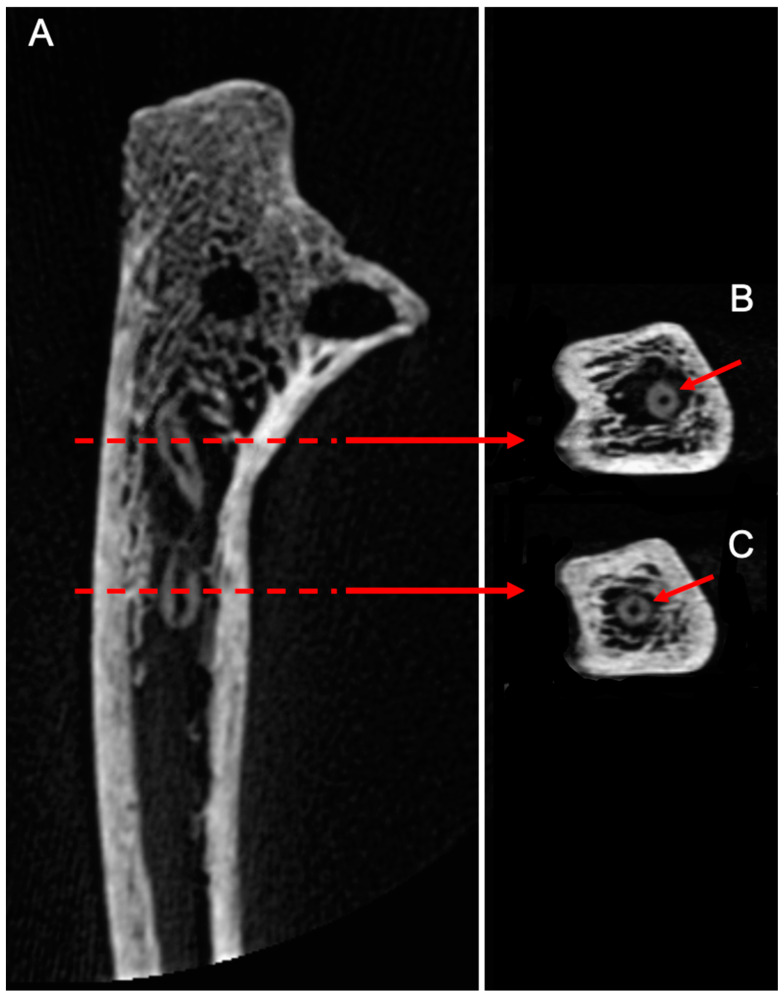
(**A**)/CBCT longitudinal section of the human ulna. X-rays allowed us to locate nymphs inside the medullary cavity, associated with trabecular bone destruction, attesting the presence of dermestids inside the bone. (**B**,**C**)/Cross-sections of the molts showing a round cuticular section with a central cavity.

**Figure 5 biology-11-01321-f005:**
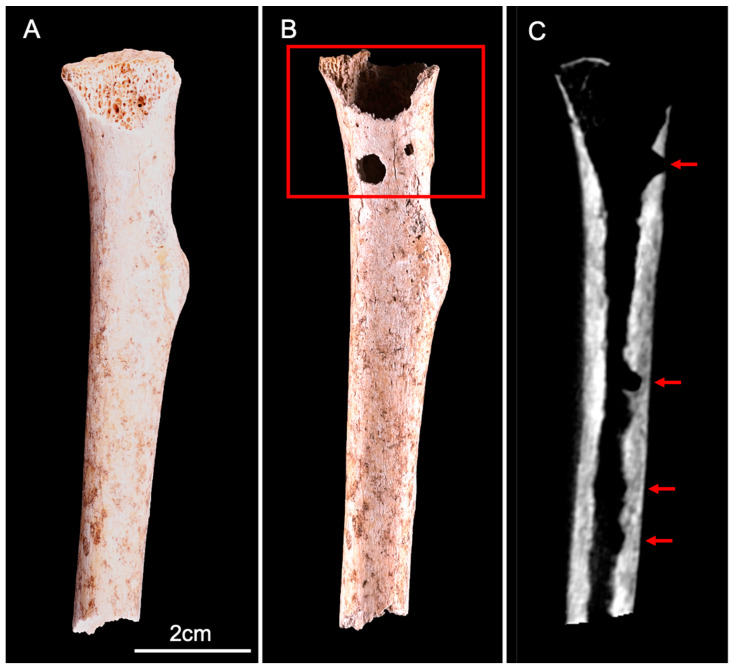
Anterior view of the human radius (proximal fragment) before (**A**) and after (**B**) the experiment. We observed an extensive destruction of the proximal epiphysis trabecular bone, associated with two cortical perforations (red square). (**C**) CBCT longitudinal section shows the trabecular destruction and thinning of the cortical inner surface (red arrows).

**Figure 6 biology-11-01321-f006:**
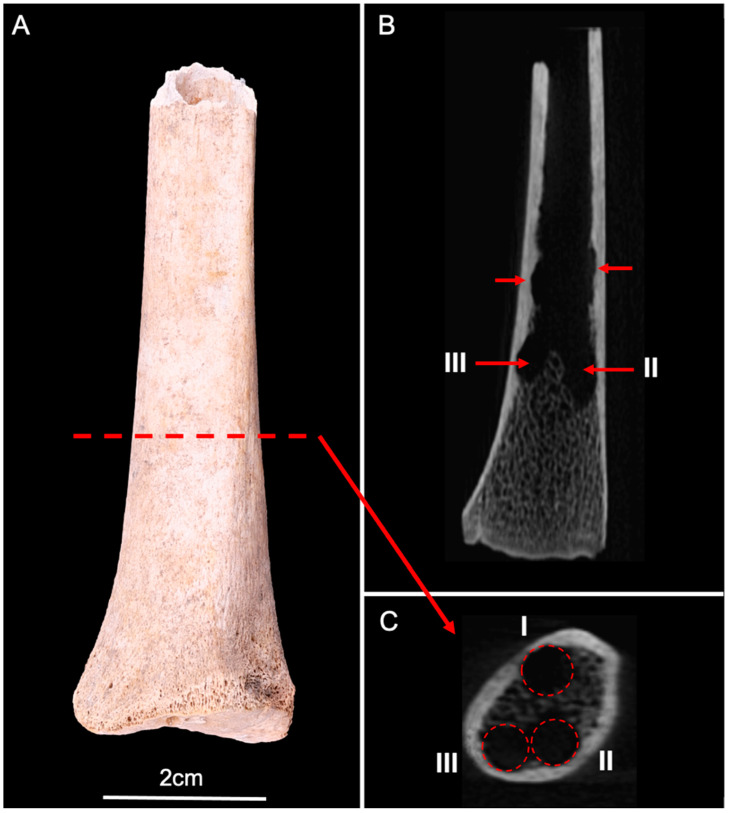
(**A**) Anterior view of the human radius (distal fragment). (**B**) CBCT longitudinal section showed several lesions of the inner surface of the cortical bone, and trabecular destructions (red arrows). (**C**) CBCT cross-section showed three circular lesions (I, II, and III) corresponding to pupation chambers in the distal epiphysis (red circles).

**Table 1 biology-11-01321-t001:** Distribution of the bones according to the initial density of *D. maculatus* larvae.

Low larval density	Adult *Bos taurus* Fresh	Juvenile *Bos taurus* Fresh	Adult Human Dry	Adult Human Dry
	Femur Distal epiphysis	Femur Distal epiphysis	Femur Proximal epiphysis	2 Lumbar vertebrae
High larval density	Adult *Bos taurus* Fresh	Juvenile *Bos taurus* Fresh	Juvenile Human Dry	Adult Human Dry
	Femoral Head	Femoral head	**Ulna, Radius (two fragments)**	**Calcaneus**

Conditions resulting in osteological lesions are displayed on grey background; bones showing deep cortical and/or trabecular perforations are reported in bold.

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
