# Peer review of "Experimental Evidence of Bone Lesions Due to Larder Beetle *Dermestes maculatus* (Coleoptera: Dermestidae)"

_biology, 2022, doi:10.3390/biology11091321_

Round 1
Reviewer 1 Report
The article deals with the study of multiple bones lesions made by dermestid beetles. The presence of Dermestid pupation chambers that can be particularly important in studing mass graves. The information presented here will add some basic data to the few literature available. The article is well written.
I only suggest few minor revisions as follow:
- Line 28 – …bones, but…
- Line 58 – remove “of” in “ten to hundreds of eggs”
- Lines 68-70 – I will rewrite as follow : “However, larvae have also been reported to sometimes cause osteological lesions ranging from grooves to furrows”
- Line 93 – add authorship for Dermestes maculatus
- Line 160 – hole instead of holes
- Panels 2A, 3A are not cited in the text
- Line 196 - E/ CBCT cross-section instead of D/ CBCT cross-section
- Line 214 – red square instead of red circle
- Line 216 - D/ CBCT cross-section instead of C/ CBCT cross-section
- For completeness, in the results section add few sentences also about bones with “negative” results
- Lines 330-331 – “...events. Information ….”
- Line 333 – fields
Reviewer 2 Report
The study for the first time uses medieval human bones for an experimental study of lesions potentially inflicted by beetles. As controls, fresh bovine bones were used. In these experiments, the beetles preferred the old human bones for digging holes but only if beetle density was high. This is supposed to help understand the chronology of mass graves disposals.
The title suggests evidence for at least a number of different larder beetle species. The study however only investigates Dermestes maculatus. The keywords also omit this detail. I think the title should reflect the fact that only one species was investigated.
Introduction
One publication by the first author of the current manuscript was cited no less than four times in the first two paragraphs of the introduction, even for very general statements. Please consider inclusion of more universal references.
According to the introduction it was previously shown, that larder beetles are capable of forming pupal chambers in human bones. How than can the aim of the study be “(…) to test the ability of dermestid larvae to dig pupal chambers in human bones (...)” (Line 89)?
Materials and Methods
Line 107: What was the purpose of the dry food supplied to the beetles during the experiments? Is this comparable to the conditions beetles would encounter in mass graves? As the aim of the study was to determine the conditions favoring a specific behavior, could food supply not be an important factor?
Line 111: Boxes for “20 larvae experiments” were not the same as for “40 larvae experiments”. Could the size of the box have additional impacts on the experiments?
Line 114: claims to have used three different types of bones. This is not correct! All bovine bones were fresh and all dry bones were human. For a real comparison, either old bovine bones or fresh human bones should additionally have been used.
As an additional control, at least one of the dry bones should have been placed in a box with cat pellets and damped cotton without larvae to see the effects of humidity to the previously probably very dry bones.
The experimental setup is not clear: eight experiments were conducted simultaneously with all available bones divided up into two boxes (Lines 120 and 122)? How would this work?
Results
Line 156: The authors observed pupae inside the cavities found on the soft parts of the fresh bones. In the dry bones, only molds could be found (Line 226) but no pupae, did I understand this correctly?
If so where did the larvae go to pupate? If I understand correctly, the experiments ended as soon as no moving larvae were visible (Line 110). Where did they end up?
Discussion
I do not understand how alterations created by D. maculatus can provide information on the taphonomic history of bone remains (Line 276-277). Please explain further.
The authors state the limited set of experiments (Line 279) and I agree: these experiments absolutely need to be replicated using comparable bones (dry and fresh from the same species) and a viable control (e.g. bones in experimental conditions without larvae).
What other taphonomic processes (Line 288) could lead to hollow bones?
How are the lesions in dry bones be “an indication of asymmetric decomposition” (Line 309)? Why would “pupation chambers suggest the presence of several corpses at different decomposition stages” (Line 312)? Sorry, this line of reasoning completely eludes me and I need more explanation.
How does the presence of pupation chambers in some bones mean that all the other bones are fresher or in different decomposition stages?
Why would “the presence of another corpse, not yet skeletonized,” (Line 325) “be necessary to explain the successful development of the feeding larvae” (Line 326) if “Dermestid beetles (Coleoptera: Dermestidae, also known as larder beetles), are necrophagous insects feeding on dry organic material” (Line 44)?
Reviewer 3 Report
The manuscript contains new and significant information to justify publication. The problem is actual, and the experimental methods are comprehensible and adequately described. The introduction is concise, the discussion is justified by the results, and the literature citation is adequate. Additionally, references are up to date and relevant.
Furthermore, the manuscript meets the publication criteria of the Biology. So, I recommend the publication, with minor revision:
1. Line 31: italics are missing in Bos taurus.
2. Lines 54-55: the first time a species appears in the text it must have the descriptor (the name of the authors and the date on which it was described).
3. Line 108: the citation rule (by numbers) was not followed “…(Gay, 1938; Guyot, 1996).”
4. Lines 157, 185, 188, 189, 207, 209, 228, 241, 245, 261: need for standardization - "figure x" vs "Figure x".
5. References also need standardization, as names of Journals are in full and abbreviated.
Reviewer 4 Report
This paper reviews an important aspect of dermestid behavior in relation to bone damage in a forensic context. This is an area understudied in the field of forensic entomology. These initial findings lay the foundation for more work to be completed in this area. Please see specific comments below.
Simple Summary: Should be larder beetle larvae rather than larder beetles larvae (line 13,18)
Abtract: Bos taurus should be italicized (line 31).
Introduction: The first time you introduce a species you should include the full genus and the authority (lines 54-55).
typo, 'nypmha' should be 'nymphs' (line 57)
dermestid beetle larvae should be used as beetles larvae is a double plural or you're trying to give it possession (beetle's larvae) (line 74).
Experimental design- why were the colonies kept in complete darkness? (line 109)
Did you let them sit for 30 days with no intrusion?
Capitalize table title (Line 146).
calcaneus may be more appropraite that calcaneum
Figure 3- why does the lighting appear so different between A&B? Is there a way to make this standard? Same with figure 4
Reviewer 5 Report
In general, the paper is good, clear and concise. Although I understand that it’s a preliminary work and the obtaining of material for its realization is scarce and difficult to achieve, some repetition with its statistical analysis is missing in the results. Even so, I believe that the results presented correspond to the objectives.
Only a few questions arise for me in the methodology: Were the different types of bones introduced into the test boxes randomly? Did they have some stratification? Were they on the same level?
I’m only going to point out some details that could improve it, not being faults in themselves or a problem for understanding if they are left as they are now. Rather they are suggestions.
On line 44 where it says “Dermestid beetles (Coleoptera: Dermestidae, also known as larder beetles), are…” it would be convenient to change it to “Dermestid beetles (Coleoptera: Dermestidae), also known as larder beetles, are…”
On line 146 change “Tab. 1” for “Table 1” and match the format of said legend with that of the figure captions.
On line 189 change “(Figure 2D)” to “(figure 2D)”.
On point 3.2 Human calcaneum, there is a figure 2 E where there is no reference in the text or in the legend of the figure.
Lastly, in section 3.3 Human ulna, the legend of the figure caption lacks a reference to image “D”. In this case, it’s included in the text of the same section.
Round 2
Reviewer 2 Report
I would like to thank the authors for the comprehensive response and the changes made to the manuscript!
The results are important as the general understanding is that larder beetles are not able to damage bones in any significant way. It should, however, be clearly stated in the text that the beetles' use for maceration of fresh bones is in no way compromised by these findings.
Furthermore, there are still some minor corrections necessary in the language of the manuscript, which I am not able to provide.
Author Response
Thank you for this comment: we have added the following sentences to the Discussion:
Larder beetles are used by natural history museums to clean skeletal remains. Their ability to quickly remove flesh from bones is also of interest for forensic anthropology laboratories. Indeed, the use of mechanicals methods to remove soft tissues can damage the surface of the bones, and simmering with detergent or bleach are not DNA-safe. However, it is necessary to ensure that dermestid larvae do not create superficial or deep lesions during the cleaning process. Together with former studies, our results indicate that D. maculatus larvae bred at low density with food ad libitum do not alter fresh bones. We however recommend to use young larvae, provide pupations substrate such as wood or polystyrene and remove the bones before larvae start to pupate.
The text has also been proofreaded and validated by a native speaker from USA.
Thank you again for your constructive feedback.
Kind regards
The authors